# *Bacillus velezensis* FX-6 suppresses the infection of *Botrytis cinerea* and increases the biomass of tomato plants

Zhaoyu Li[1]*, Jiajia Li[1], Mei Yu[1], Peter Quandahor[2], Tian Tian[1], Tong Shen[1]*

**1** School of Chemistry and Chemical Engineering, Lanzhou Jiaotong University, Lanzhou, Gansu Province, China, **2** CSIR-Savanna Agricultural Research Institute, Tamale, Ghana

* lizy@mail.lzjtu.cn (ZL); shentong@mail.lzjtu.cn (TS)

**Data Availability Statement:** All relevant data are within the paper and its Supporting Information files.

## Abstract

*Botrytis cinerea* causing tomato gray mold is a major cause of economic loss in tomato production. It is urgent and necessary to seek an effective and environmentally friendly control strategy to control tomato grey mold disease. In this study, *Bacillus velezensis* FX-6 isolated from the rhizosphere of plants displayed significant inhibitory ability against *B. cinerea* and could promote tomato plant growth. FX-6 could effectively inhibit the growth of *Botrytis cinerea* mycelium in vitro and in vivo, and the inhibitory rate in vitro could reach 78.63%. According to morphological observations and phylogenetic trees based on sequences of the 16S rDNA and gyrA (DNA gyrase subunit A) genes, the strain FX-6 was identified as *Bacillus velezensis*. In addition, *B. velezensis* FX-6 showed antagonistic activity against seven phytopathogens, this indicated that FX-6 had broad-spectrum biocontrol activity. We also found that FX-6 fermentation broth had the strongest antagonistic activity against *B. cinerea* when the culture time was 72 hours, and the inhibition rate was 76.27%. The growth promotion test revealed that strain FX-6 significantly promoted tomato seed germination and seedling growth. Further deeply study on growth-promoting mechanism indicated that the FX-6 produced IAA and siderophore, and had ACC deaminase activity. The trait of significant biological control activity and growth promoting effect on tomato imply that *B. velezensis* FX-6 has the potential to be used as a biocontrol agent for tomato gray mold management.

## Introduction

Tomato is an important vegetable grown in quite a few countries due to its high nutritional value, and China is one of the countries with the largest cultivated area worldwide [1]. *Botrytis cinerea* is the second largest most common plant pathogenic fungi globally [2], which causes gray mold diseases. *B. cinerea* attacks over 200 crops hosts, such as tomato, strawberry, grapes, and pepper [3]. Tomato gray mold disease contributed to a yield loss of approximately 20–30%, and decreased production by 50–60%, under severe conditions [1]. To date, chemical control based on the application of chemical fungicides has remained a effective way to manage tomato grey mold, due to the lack of cultivars with resistance. However, the extensive use

**Funding:** This research was supported by the Young Scholars Science Foundation of Lanzhou Jiaotong University under grant number 2019030, National Natural Science Foundation of China under grant number 201967015, and Key Research and Development Program of Gansu Province under grant number 21YF5FA059. The funders had no role in study design, data collection and analysis, decision to publish, or preparation of the manuscript.

**Competing interests:** The authors have declared that no competing interests exist.

of chemical fungicides can increase pathogens' resistance to chemicals. Significantly, *B. cinerea* is recognized as a high-risk pathogen for acquired resistance to fungicides (Fungicide Resistance Action Committee: http://www.frac.info), and researchers found that the resistance rate of *B. cinerea* to carbendazim and procymidone has reached 100% in some areas of China [4]. Moreover, long-term use of fungicides can bring many disadvantages, including leads to environmental contamination, biodiversity loss and human health risks [5, 6]. Biological control is a safe and reasonably effective alternative method for reducing the use of chemicals, and has received wide attention for plant disease control. The use of biological control method has been proposed as the alternative method due to it is considered safe and environmentally friendly. Most of these microorganisms accumulate metabolites that not only inhibit plant pathogens, but also accelerate the growth of plants. There have been several reports on the use of microorganisms for the control of tomato gray mold, such as *Pseudomonas* [7], *Bacillus*, *Streptomyces* [8], *Trichoderma* [9, 10], and yeast [11–13]. In particular, *Bacillus* strains have received a huge attention as biocontrol agents due to their ability to produce a wide range of broad-spectrum antibiotics and form endospores, which allows them to have a long shelf life as a commercial product [14]. At present, the most reported *Bacillus* species used to control tomato gray mold are: *Bacillus subtilis* [15, 16], *Bacillus amyloliquefaciens* [17, 18], *Bacillus cabrialesii* [19], and *Bacillus licheniformis* [3, 20]. However, there are few reports on the control of tomato gray mold by *B. velezensis*.

Most *Bacillus* strains isolated from tomato or other plant rhizosphere soil had not only antagonistic but also growth-promoting activity. Soil bacteria that promote plant growth are referred to as plant growth promoting rhizosphere bacteria (PGPR). PGPR promotes plant growth through a variety of mechanisms, such as the production of plant growth regulators including indole acetic acid (IAA); biological nitrogen fixation; solubilization of soil phosphorus compounds; secretory siderophores; and resistance induction [21]. The main purposes of studying PGPR are to develop strains with PGPR characteristics and the functions that can be used as biological fertilizers and biological control agents in agricultural production.

This study aimed to isolate biocontrol bacteria from the rhizosphere soil of plants and screen them against tomato gray mold. The research studied the antagonistic activity of FX-6 against *B. cinerea* and the growth-promoting effect of FX-6 suspension on tomato seeds and seedlings. Moreover, the antifungal spectrum, optimal fermentation time and plant growth-promotion substance of FX-6 were also evaluated. We found that *B. velezensis* FX-6 could act as a potential and efficient biocontrol agent, which not only can significantly control tomato gray mold disease, but also promote the tomato plant growth. These findings demonstrate the feasibility of *B. velezensis* FX-6 to be applied in practice for tomato disease control.

## Materials and methods

### Fungi, plants and biopesticide

A total of eight pathogenic fungi were used to determine the antagonistic activity of strain F3A, among which *Valsa mali* ACCC37712 and *Alternaria mali* ACCC36255 were obtained from Agricultural Culture Collection of China (ACCC, http://www.accc.org.cn). Other six phytopathogenic fungi were provided by Plant pathology laboratory in Institute of Plant Protection of Gansu Academy of Agricultural Sciences, they were isolated from tomato, watermelon, eggplant, potato, apple and Chinese wolfberry tissues in Lanzhou City, Gansu province, China, including: *Botrytis cinerea*, *Fusarium oxysporum*, *Sclerotinia sclerotiorum*, *Rhizoctonia solanikuhn*, *Colletotrichum gloeosporioides* and *Colletotrichum acutatum*. In particular, the *B. cinerea* was indicator fungi.

In this study, we used one kind of tomato cultivar (Lucas) as a test plant in the greenhouse trials, which was bought from Shandong Boya Agricultural Technology Co. Ltd. The variety was susceptible to *B. cinerea*.

*Bacillus subtilis* wettable powder (100 billion/g) were produced by Jiangxi Zhengbang Crop Protection Co., Ltd (Jiujiang China), a biopesticide, was used as a control for antagonistic activity determination.

## Isolation and screening of antagonistic bacterium

Soil samples were collected from the bank of Eling Lake at an altitude of 4272 meters in the west of Maduo County (3808' N 9707' E), Qinghai Province. The soil samples were transported to the laboratory under controlled temperature conditions. The isolation of antagonistic bacteria was performed as described by Li et al. with modifications [22]. Generally, approximately 10 g soil sample was suspended in a 250 mL flask with 100 mL sterilized water and was shaken using a rotary shaker at 28 C, 180 rpm for 30 min, the sample was taken out and then it was stayed still for 30 min. After stratification, 1 mL of supernatant was then diluted to $10^{-5}$, $10^{-6}$ and $10^{-7}$ with sterilized water, and 100 µl of each diluent were spread on LB plates, incubated at 28 C for 2 days. All bacteria were isolated from single colony on LB plates, and stored at -4 C for later use.

The plate confrontation culture method was used to screen the antagonistic bacteria [23]. A mycelia plug (0.5 cm) containing *B. cinerea* mycelium was excised with a hole punch from the edge of a 5-day-old fungal culture and placed on the center of a new PDA plate. The isolated bacteria were inoculated on the plates approximately 2.5 cm away from the plug of *B. cinerea* at four points at the same distance. PDA plates without inoculation of isolated bacteria were used as control. Three replicates were used per isolate and the experiment was repeated three times. When the plates of the control were fully covered with the fungal growth, cultured at 25 C for about 6 days, the inhibition zones were observed. Bacteria that showed inhibition zones were thought to be antagonistic to *B. cinerea*. The inhibitory rate was used to express antagonistic activity, strains with a high inhibitory rate had a high antagonistic activity.

The inhibitory rate was calculated as follows: the inhibition rate (%) = $[(A1 - A2)/A1] \times 100\%$
where $A1$ = colony diameter of control group, and $A2$ = colony diameter of treatment

## Identification of bacterial isolate FX-6

Morphological identification: the cultured FX-6 bacterial suspension was streaked on LB medium plate, placed at 28 C for 24 h. Colony characteristics, such as shape, size, color, and transparency, were observed. A single colony was taken and spread evenly on a glass slide and then dry-fixed and Gram-stained for 1 min. Next, the morphology, size, and spore characteristics of strain FX-6 was observed under a microscope.

Sequence analysis of 16S rDNA and gyrA genes: total DNA of strain FX-6 was extracted by genomic DNA isolation kit (Sangon, China) according to the manufacturer's instructions. The 16S rDNA gene fragments were amplified by using bacterial universal primers 27F (5'-AGA GTT TGA TCC TGG CTC AG-3') and 1492R (5'-ACGGCT ACC TTG TTA CGA CTT-3'), the gyrA gene fragments were amplified by using primers gyrA-F (5'-CAG TCA GGA AAT GCG TAC GTC C-3') and gyrA-R (5'-CAA GGT AAT GCT CCA GGC ATT GCT-3') [24]. The PCR reactions were performed in 25 µL solution containing 12.5µL 2×Taq PCR MasterMix (Sangon, China), 0.5 µL each primer, 0.5 µL DNA template, and ddH$_2$O in order to make up the remaining volume. The PCR thermal cycle profile was as follows: denaturation at 94 C for 5 min; 30 cycles of 45 s at 94 C, 45 s at 54 C, 1 min 30 s at 72 C; final extension at 72 C for 7 min, and cooling at 4 C.

The PCR products were purified by PCR product purification kit (Sangon, China), and were handed over to Beijing Huada Gene Company for sequencing analysis. Sequence analysis of 16S rDNA was performed with the program EZBioCloud (https://www.ezbiocloud.net/). The gyrA gene sequence was compared with those of different species deposited in the National Center for Biotechnology Information database (http://www.ncbi.nlm. Nih.gov). Phylogenetic and molecular evolutionary analyses were carried out using MEGA version 6.0. The phylogenetic trees were constructed by using the neighbor-joining method with bootstrap analysis of 1000 replicates.

## Determination of inhibition spectrum of FX-6

The antagonistic activity of strain FX-6 against seven other pathogenic fungi, namely *V. mali*, *A. mali*, *F. oxysporum*, *S. sclerotiorum*, *R. solanikuhn*, *C. gloeosporioides* and *C. acutatum*, was determined by using the plate confrontation culture method with minor modifications [23]. Mycelial plugs with a diameter of 5 mm were cut from the edge of different fungal cultures and placed on the center of PDA plates. The FX-6 was inoculated on the plates approximately 2.5 cm away from the plug of different fungi at four points at the same distance, which was then cultured at 25 C. PDA plates without inoculation of FX-6 were used as control. The fungal colony diameters of the plate inoculated with FX-6 were measured when the fungal mycelia of control reached the edges of the plate. The inhibition rate of strain FX-6 against different fungi was calculated. Each treatment was performed with three plates.

## Determination of fermentation time of FX-6

The effect of fermentation time of strain FX-6 on the antifungal activity against *B. cinerea* was determined. First, strain FX-6 was cultured in LB liquid media until its OD600 reached 2.0, and then inoculated in 100 mL of LB liquid medium with an initial inoculum volume of 1% at 28 C and 180 r/min for 24 h, 36 h, 48 h, 60 h, 72 h, 84 h and 96 h, respectively. The fermentation broth at different times was centrifuged at 8000 r/min for 20 min, and the supernatant at different fermentation time was collected. The supernatant was filtered through a filter membrane to obtain the sterile filtrate, and then added to PDA medium at a concentration of 10%, with the control plate containing the equal amount of sterile water. A 0.5 cm diameter *B. cinerea* mycelial plug was inoculated on the plate and cultured at 25 C for 6 days, and then the inhibition rate was determined by measuring the growing diameter of the *B. cinerea*. The inhibition rate was calculated as described previously. Each treatment was performed with three plates.

## Bioassays in plant tissue

With minor modifications, we used the assay described by Zheng et al. [25]. The pathogen inoculation bioassay was performed at the 6-leaf stage of tomato plant. In vitro tomato leaves with same size were selected and washed with sterilized water for 3 times and dried with paper towel. The leaves were then soaked in fermentation broth of strain FX-6 and *Bacillus subtilis* bacterial suspension ($1.0×10^9$ cfu/mL) for 10 seconds before being dried. The center of the leaves were inoculated with 0.5 cm of *B. cinerea* cake, the leaves were then placed in petri dishes (20 cm in diameter) with wet filter paper. In addition, the petiole of each leaf was wrapped with soaked cotton. While the leaves soaked in water served as a control, each treatment contained three leaves. The treated leaves were then cultured at 25 C, and the disease development was observed on the 2st, 3rd, and 5th days. The experiments were performed three times.

## Estimation of plant growth-promotion substance

In order to detect plant growth-promoting traits, some of Bacteria FX-6's biological functions were determined, including: phosphate solubilization, siderophore, indole-3-acetic acid (IAA) production and 1-aminocyclopropane-1-carboxylate (ACC) deaminase activity.

Siderophore production was estimated qualitatively by inoculating the bacteria FX-6 on CAS detection medium at 28 C for 14 days. At the end of the incubation period, transparent zone around the colony was observed, indicating that the strain FX-6 had the ability to produce siderophores.

The IAA production was detected by the Salkawski reagent assay as described by Xu et al. [14]. Approximately 1 mL of strain FX-6 fermentation broth was inoculated into 100 mL of King's B broth containing 100 mg/L L-tryptophan and without L-tryptophan, respectively, and incubated at 28 C, 180 r/min for 72 h. The King's B broth was inoculated with the same amount of sterile water was used as blank control. The fully grown culture was centrifuged for 10 min at 12,000 r/min. Then the supernatant (2 mL) was mixed with 4 mL of Salkowski reagent (12 g FeCl3 was dissolved in 300 mL distilled water, and add 429.7 mL of 98% $H_2SO_4$ slowly, after cooling, constant volume to 1 L). When the mixed solution appears pink, it indicated the production of IAA. Optical density (530 nm) was measured with a spectrophotometer after 30 min incubation. The concentration of IAA was estimated using a standard graph spiked with commercial IAA.

The estimation of P solubilization was performed by inoculating the bacteria FX-6 on organophosphorus solid medium (dextrose 10 g/L, $(NH_4)$ $SO_4$ 0.5 g/L, NaCl 0.3 g/L, KCl 0.3 g/L, FeSO_4 0.03 g/L, $MgSO_4 \cdot 7H_2O$ 0.3 g/L, $MnSO_4 \cdot 4H_2O$ 0.03 g/L, $CaCO_3$ 5 g/L, lecithin 0.2g /L, and agar 20 g/L) and inorganic phosphorus solid medium (dextrose 10 g/L, $(NH_4)$ $SO_4$ 0.5 g/L, NaCl 0.3 g/L, KCl 0.3 g/L, FeSO_4 0.03 g/L, $MgSO_4 \cdot 7H_2O$ 0.3 g/L, $MnSO_4 \cdot 4H_2O$ 0.03 g/L, $Ca_3(PO_4)$ 5 g/L, and agar 20 g/L) for 14 days at 28C. We checked the transparent zone around the colony at the end of the incubation period to ensure that the strain has the ability to dissolve organophosphorus and inorganic phosphorus.

ACC deaminase activity was determined as described by Qin et al. [26]. The bacteria FX-6 were inoculated into a sterilized Dworkin and Foster (DF) salt minimal medium containing 3 mmol/L ACC as the only nitrogen source. When the medium is turbid after 5 days of shaking culture at 28 C and 180 r/min, it indicates that the strain FX-6 has ACC deaminase activity.

## Growth-promoting effect of FX-6 on tomato seeds and seedlings

The same-size tomato seeds were selected and surface sterilized before being immersed in 2% sodium hypochlorite (v/v) for 15 min, 75% alcohol for 30 s, and then washing three times in sterile water. The sterilized seeds were soaked for 30 mins in vario dilutions of strain FX-6 fermentation broth and cultured at 25 C under constant temperature and humidity. After 6 days, the germination rate and average radicle length were recorded and the seeds were soaked in sterile water used as blank control. The dilution factors were 1, 10, 50, 100, and 200, respectively, with 10 seeds were selected for each treatment, and each treatment was repeated three times.

The treated seeds were then sown in pots. The culture of *B. velezensis* FX-6 with different dilution times (10, 50, 100, and 200) were root-irrigated with an amount of 20 ml per individual plant when the seedlings grew to 6 leaves status, the sterile water was used to serve as a control. The inoculated tomato seedlings were maintained in the greenhouse at 25–30 C. The plant height, stem thickness, main root length, and wet weight of tomato seedlings were measured after 45 days.

The equation to calculate the germination rate was: Germination rate (%) = (B1/B2) × 100% where B1 = the number of germinated seeds, and B2 = the total number of seeds.

## Statistical analysis

Data from the experiment was analyzed by one-way ANOVA using SPSS 24.0 software, and represented as the mean ± SD. Significance of mean differences was determined using the Duncan's test, with a significance level of 5% (P = 0.05).

## Results

### Isolation of antagonistic bacteria

A total of 50 bacterial strains were isolated and used in the antagonistic screening as a pool. After screening for antifungal activity, seven bacteria with varying degrees of antagonistic activity against *B. cinerea* were obtained (Fig 1). Among them, strain 6 had the strongest inhibitory effect on *B. cinerea* mycelial growth, with a 78.63% inhibitory rate (Fig 2). Moreover, the antagonistic activity of FX-6 did not significantly decrease from the first generation to the tenth generation (data not shown). As a result, strain 6 was chosen for further investigation and named FX-6.

Error bars display the standard deviation of three replicates performed for each treatment. Different lowercase letters indicate significant difference at P<0.05 level.

### Identification of the antagonistic bacterium FX-6

The strain FX-6 was gram-positive, and appeared under the microscope as small rods with spores. After cultured at 28 C for 3 days on LB medium plate, strain FX-6 had a single colony diameter of about 3 mm, with milky white, irregular edges, wrinkled and upward protruding surfaces.

The 16S rDNA and gyrA genes of strain FX-6 phylogenetic analysis revealed that the bacterium was closely related to *B. velezensis* (Fig 3). Therefore, strain FX-6 was identified as *B. velezensis*. The GenBank accession numbers of 16S rDNA and gyrA genes of strain FX-6 were ON209683 and ON263393, respectively.

### Inhibitory spectrum of strain FX-6

The strain FX-6 showed inhibitory abilities against all tested fungal strains (Fig 4). However, the antifungal effects of each strain were different. Of the seven phytopathogens, *S. sclerotiorum* was shown to be the most sensitive to FX-6, and the inhibition rate was87.24%. FX-6 also inhibited *C. gloeosporioides* and *A. mali*, and the inhibition rates were 46.26% and 36.67%. Other plant pathogenic fungi were inhibited at rates ranging from 50% to 70%. The results demonstrated that strain FX-6 had broad-spectrum antifungal activity.

Error bars display the standard deviation of three replicates performed for each treatment. Different lowercase letters indicate significant difference at P<0.05 level.

### Optimum fermentation time of FX-6

The supernatant of strain FX-6 at different fermentation time was collected andthe antagonistic activity against *B. cinerea* was determined. As shown in Fig 5, when the culture time increased from 24 h to 72 h, the inhibition rate increased. When the fermentation time was 72h, the inhibition rate reached the maximum and the inhibition rate was 76.27%. After more than 72 h culture, the inhibition rate gradually decreased, and the inhibition rates at 84 h and 96 h were 44.31% and 44.53%, respectively. The results showed that FX-6 had the greatest

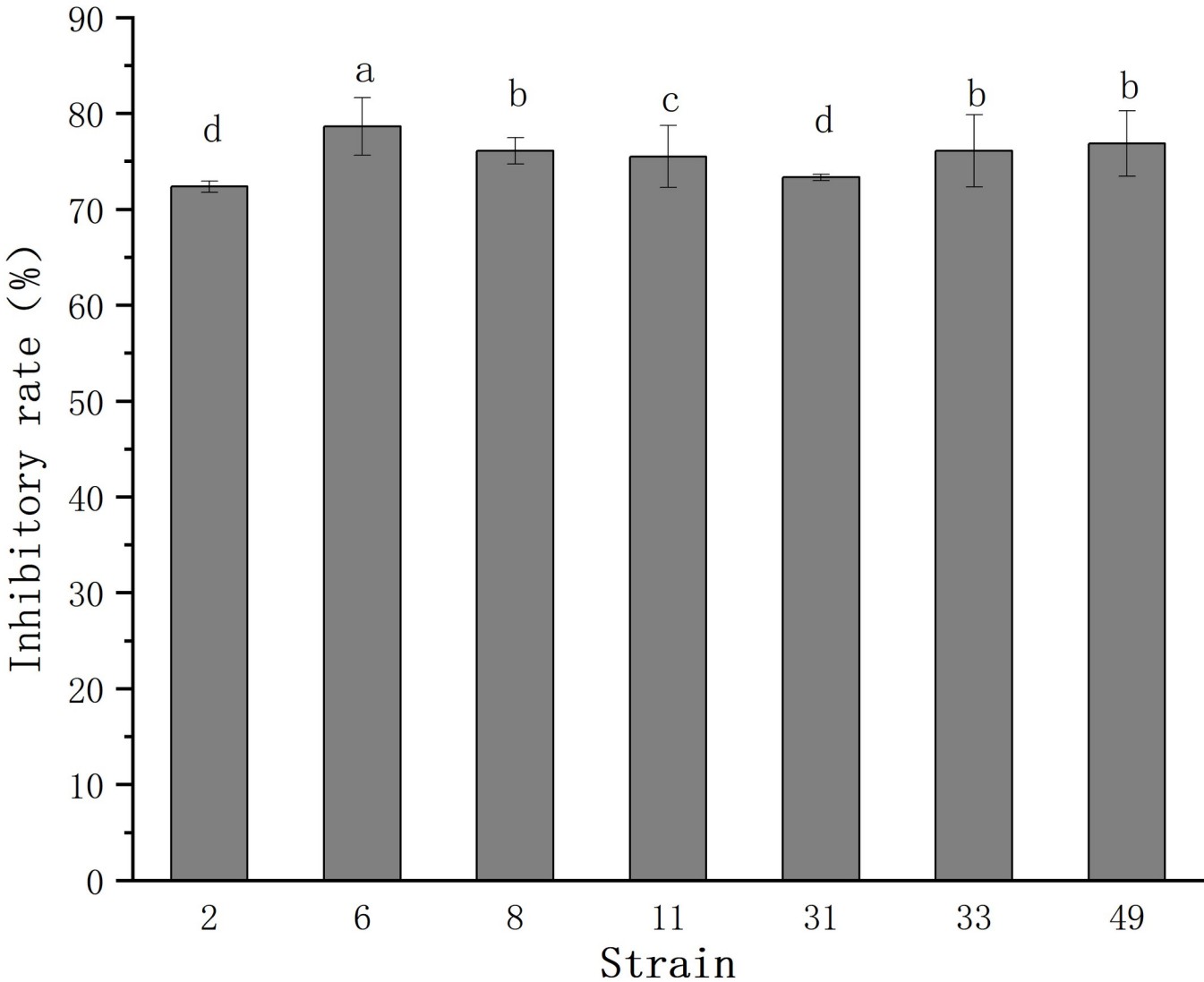

**Fig 1. Inhibitory effects of seven bacterial strains against *B. cinerea* in vitro.**

inhibitory effect on *B. cinerea*, when the culture time was 72h. As a result, we chose the fermentation time of 72 hours for subsequent test.

Error bars display the standard deviation of three replicates performed for each treatment. Different lowercase letters indicate significant difference at P<0.05 level.

## FX-6 protects tomato leaves from *B. cinerea* invasion

As shown in Fig 6, two days after inoculation with *B. cinerea*, visible illness spots appeared in the leaves of control tomatoes. On the 5th day, all of the control leaves developed disease and showed severe symptoms, including disease spots and leaf yellowing. The leaves treated with *B. subtilis* showed slight spots on the second day and significant expansion on the fifth day after inoculation with fungus cake. However, the disease spot development on the leaves treated with strain FX-6 was significantly limited, even on the fifth day, it expanded slightly. We could found that the biocontrol effect of FX-6 was obviously better than that of *Bacillus*

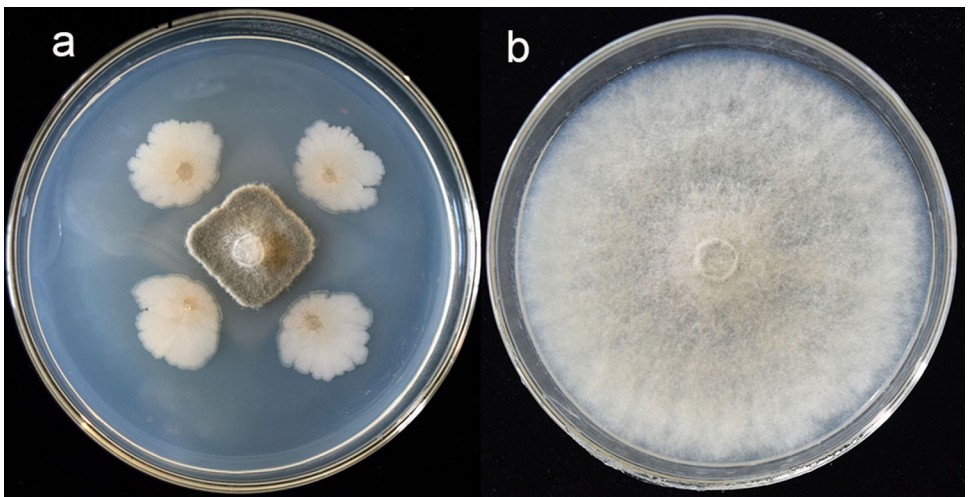

**Fig 2. Antagonistic effect of strain FX-6 against *B. cinerea*.** (a) Treated with Strain FX-6. (b) Control.

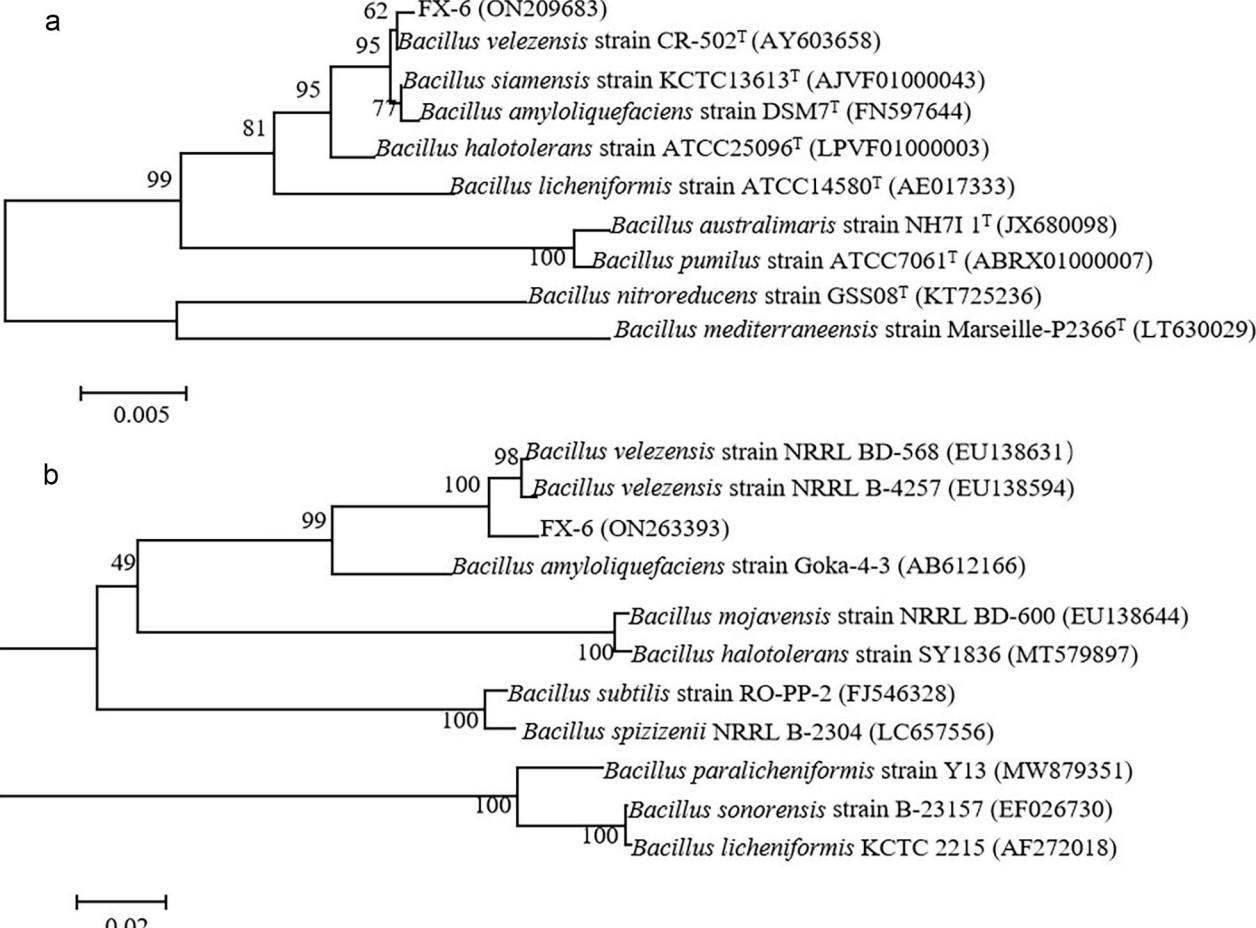

**Fig 3. Neighbor-joining tree of strain FX-6 and related species retrieved from GenBank.** (a) Phylogenetic analysis based on the 16S rDNA sequences. (b) Phylogenetic analysis based on the gyrA gene sequences. The superscript "T" indicates the type strain. The numbers at the tree branch points indicate the percent bootstrap support for 1000 iterations.

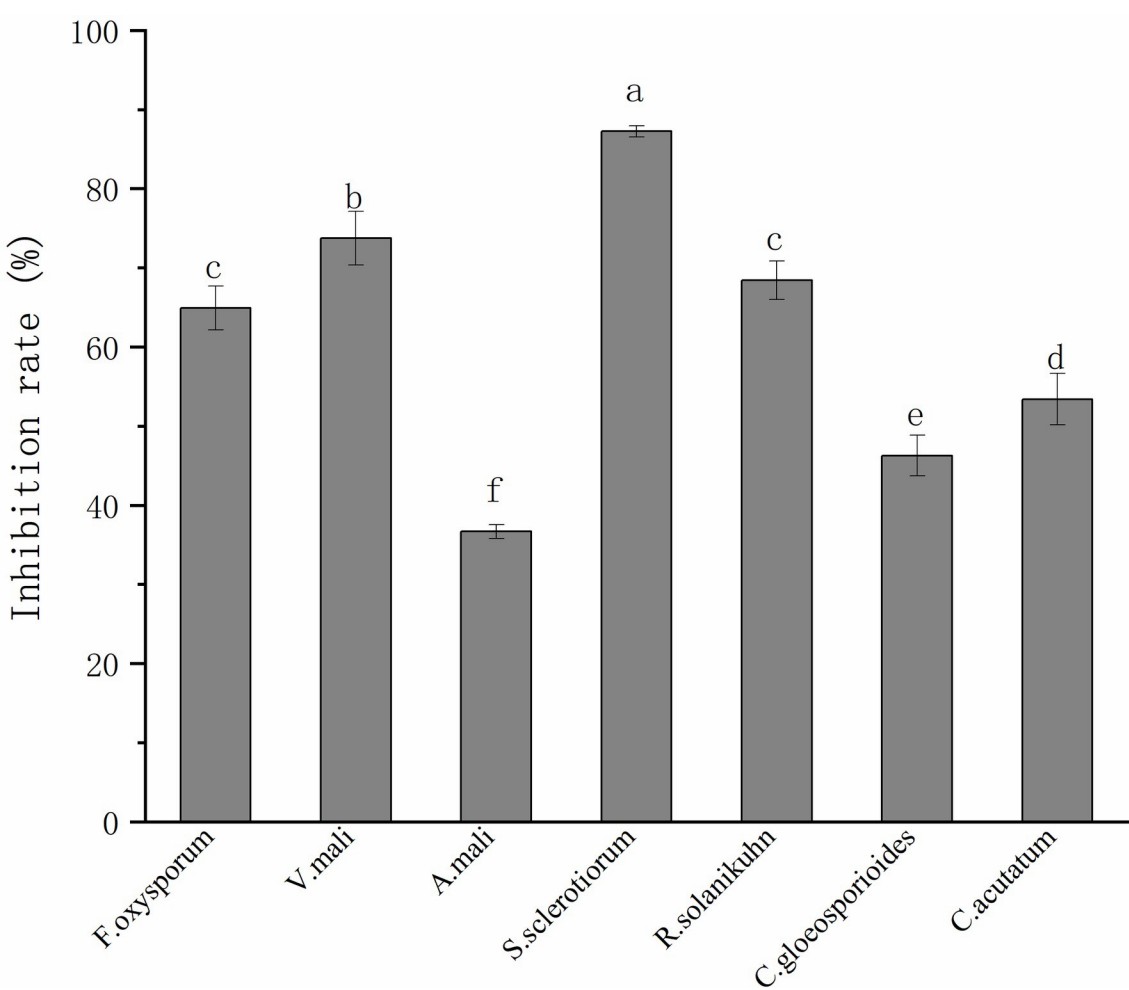

**Fig 4. Inhibition rates of strain FX-6 against seven plant pathogenic fungi.**

*subtilis* wettable powder. As a result, its indicated that strain FX-6 could effectively restrain the development of tomato gray mould disease.

## Estimation of growth promoting substances in strain FX-6

The FX-6's plant growth promotion feature was identified. The FX-6 produced IAA (Fig 7A) and siderophore (Fig 7C), as well as ACC deaminase activity (Fig 7B), but it had no phosphate solubilization, whether inorganic (Fig 7D) or organophosphorus (Fig 7E). In addition, a standard curve was made according to the standard sample, that is, Y = 0.117x+0.017 ($R^2$ = 0.997), where X represented the amount of indoleacetic acid and Y represented the absorbance value at wavelength 530 nm. The $OD_{530}$ value of colorimetric solution was 0.204 in a culture medium containing L-tryptophan, and 0.205 in tryptophan-free culture medium. The amount of IAA in Kings culture medium containing L-tryptophan was 1.598 mg/L, and the amount of IAA in Kings culture medium without L-tryptophan was 1.607 mg/L, according to the standard curve. There was no significant difference in IAA content between them, indicating that strain FX-6's synthesis of indoleacetic acid was independent of L-tryptophan.

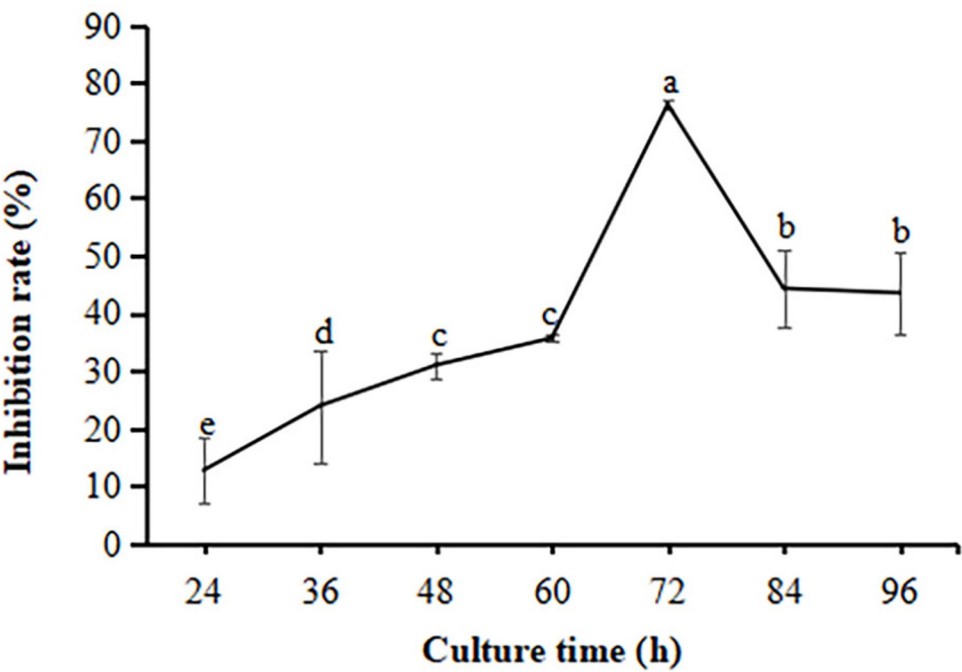

**Fig 5. Effect of culture time of strain FX-6 on inhibition rate of *B. cinerea*.**

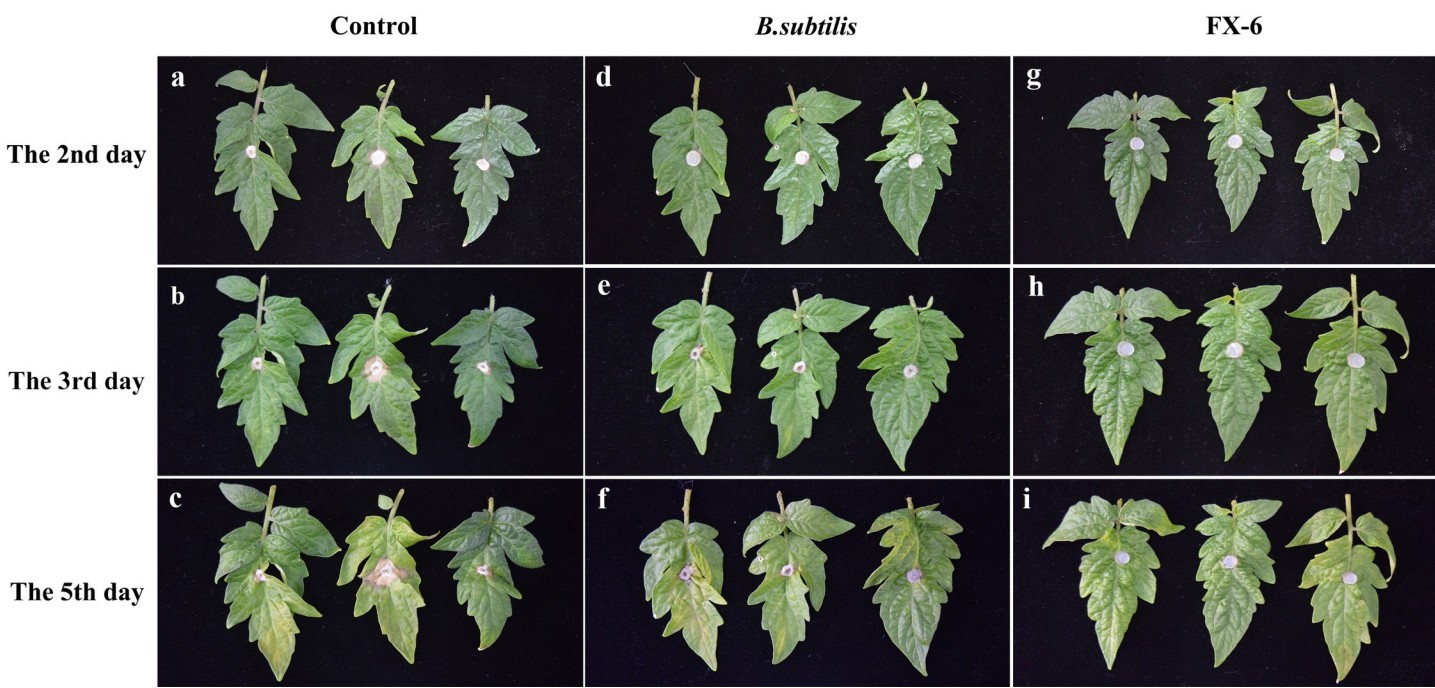

**Fig 6. Inhibition effect of strain FX-6 towards *B. cinerea* on tomato leaves at 2,3 and 5 days.** (a-c) Control. (d-f) Treated with *B. subtilis*. (g-i) Treated with strain FX-6.

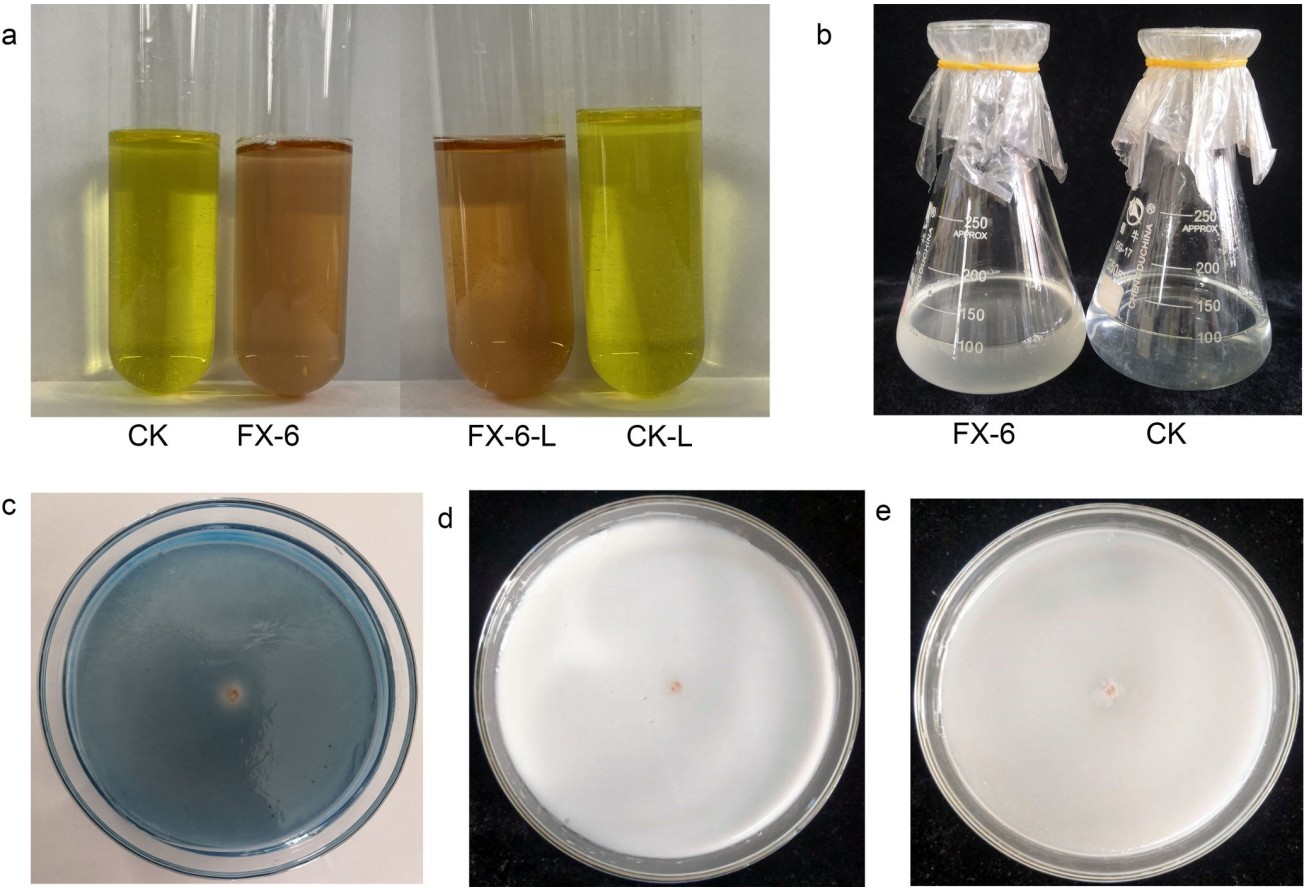

**Fig 7. Plant growth promotion attributes of strain FX-6.** (a) Test of IAA (L means medium containing L-tryptophan). (b) Test of ACC ammonia enzyme activity. (c) Test of siderophore production. (d) Test of inorganic phosphorus solubilization. (e) Test of organophosphorus solubilization.

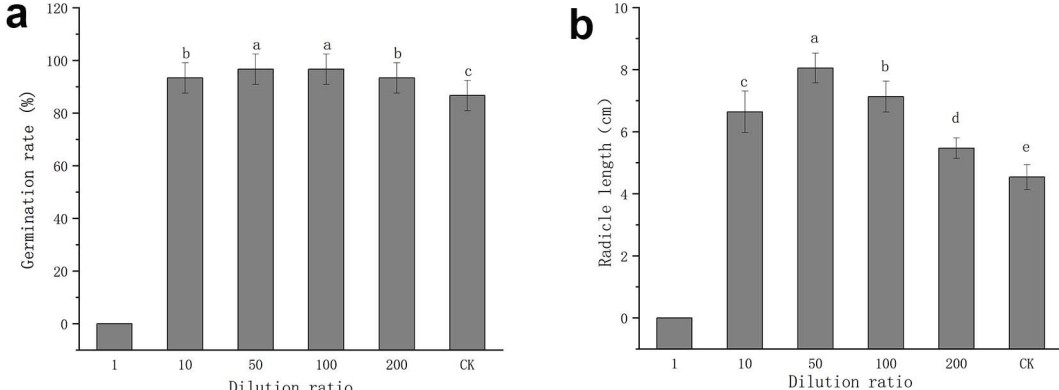

**Fig 8. Effect of different strain FX-6 dilutions on tomato seeds.** (a) Germination rate of tomato seeds. (b) Radicle length of tomato seeds. Error bars display the standard deviation of three replicates performed for each treatment. Different lowercase letters indicate significant difference at $P < 0.05$ level.

### Growth-promoting effect of FX-6

The fermentation broth of FX-6 with different dilution could promote tomato seed germination, and the germination rates were all above 90%, which was higher than that of the water control (Fig 8A). However, when the dilution ratio was 1, the seed germination rate was 0. In addition, radicle length of the treatment group was longer than that of the water control (Fig 8B). It indicated that FX-6 could promote tomato seed germination, however, the FX-6 fermentation broth must be diluted when in use.

After sowing tomato seeds treated with different dilution times of FX-6 fermentation broth, we measured the morphological indicators of tomato seedlings with a vernier caliper and a balance. The results showed that the different diluents of FX-6 had apparent growth-promoting effects for plant height, stem thickness, taproot length, and fresh weight, compared to the control group (Figs 9 and 10). Especially, when the dilution ratio of FX-6 was 100, the growth promoting effect was the strongest, the plant height, stem thickness, taproot length, and fresh weight were 1.57,1.31, 1.26 and 1.69 times of the water control, respectively.

## Discussion

The use of biocontrol agents to control tomato gray mold disease is a safer and more effective way to reduce indiscriminate application of chemicals in agricultural production. To obtain a good biocontrol agent against *B. cinerea*, 50 bacteria were isolated from rhizosphere samples, and we screened 7 strains with high inhibitory effect on *B. cinerea* (Fig 1). Among them, we found that the strain FX-6 had the strongest and most stable effect on *B. cinerea*. Therefore, the strain FX-6 was used in following experiment. According to morphological characteristics and the genes of 16S rDNA and gyrA, strain FX-6 was identified as *B. velezensis*. FX-6 could inhibit the growth of *B. cinerea* mycelium in vitro, and the inhibitory rate could reach 78.63% (Fig 2). Furthermore, we determined the biocontrol effects of FX-6 on tomato leaves, and the disease spot development was significantly limited. Even on the fifth day after inoculation with *B. cinerea*, the disease spot expanded slightly (Fig 6). These results significantly showed that the FX-6 could effectively inhibit *B. cinerea*.

*B. velezensis* is a new species discovered and named by Ruiz-García et al. [27] in 2015, it was previously thought to be a subspecies of *B. subtilis*, which is also closely related to *B. amyloliquefaciens*. In recent years, there are many reports about *B. velezensis* as biocontrol agent for a variety of plant pathogens. For instance, Chen et al. [28] reported that *B. velezensis* LM2303 had strong antagonist activity against *Fusarium* head blight; the study of Ben Gharsa et al. [29] describes that *B. velezensis* strain MBY2 can be used as a biological agent for crown gall disease management; Chen et al. [30] reported that *B. velezensis* ZW10 has the potential to be developed into a biopesticide for the biocontrol of rice blast, because its antifungal substances had the same control effect as carbendazim in the field experiment; Yan et al. [31] reported that *B. velezensis* SDTB038 had a good antagonistic effect against *Phytophthora infestans* in potato. Dong et al. [32] reported that *B. velezensis* SC60 exhibited a broad antifungal spectrum against soil-borne plant pathogens. In vitro antagonism test showed that the isolate strain FX-6 had a broad antifungal spectrum, inhibiting *V. mali*, *A. mali*, *F. oxysporum*, *S. sclerotiorum*, *R. solanikuhn*, *C. gloeosporioides* and *C. acutatum* (Fig 4), which is consistent with these preceding reports. In addition, the antagonist activity of FX-6 against *S. sclerotiorum* was even higher than that against *B. cinerea*, and the inhibition rate was 87.24%. Therefore, the results showed that FX-6 had broad-spectrum antifungal activity.

Some studies have shown that *B. velezensis* is a potential biocontrol agent due to its ability to produce antibacterial secondary metabolites such as antibiotic lipopeptides (surfactin, fengycin, and bacillomycin-D), polyketides (macrolactin, bacillaene, difficidin or oxydifficidin),

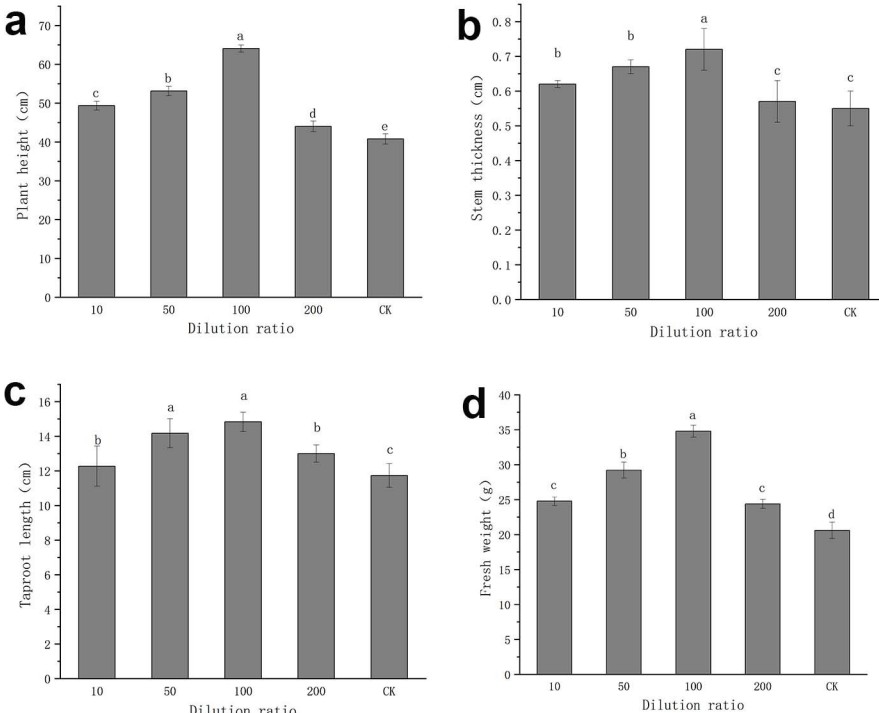

**Fig 9. Effect of different strain FX-6 dilutions on morphological indexes of tomato seedlings.** (a) Plant height of tomato seedlings. (b) Stem thickness of tomato seedlings. (c) Taproot length of tomato seedlings. (d) Fresh weight of tomato seedlings. Error bars display the standard deviation of three replicates performed for each treatment. Different lowercase letters indicate significant difference at P<0.05 level.

and peptides (plantazolicin, amylocyclicin, and bacily [33, 34]. Moreover, the genomic analysis of *B. velezensis* also revealed strain-specific clusters of genes involved in secondary metabolite biosynthesis, these gene clusters are important for pathogen suppression as well as plant growth promotion. In particular, *B. velezensis* has a high genetic capacity for cyclic lipopeptide and polyketide synthesis [35, 36]. In this study, we measured the inhibitory effect of FX-6 on *B. cinerea* under the different time to select the optimum culture time. We found that the supernatant had the maximum inhibition rate, when the culture time was 72h (Fig 5). This

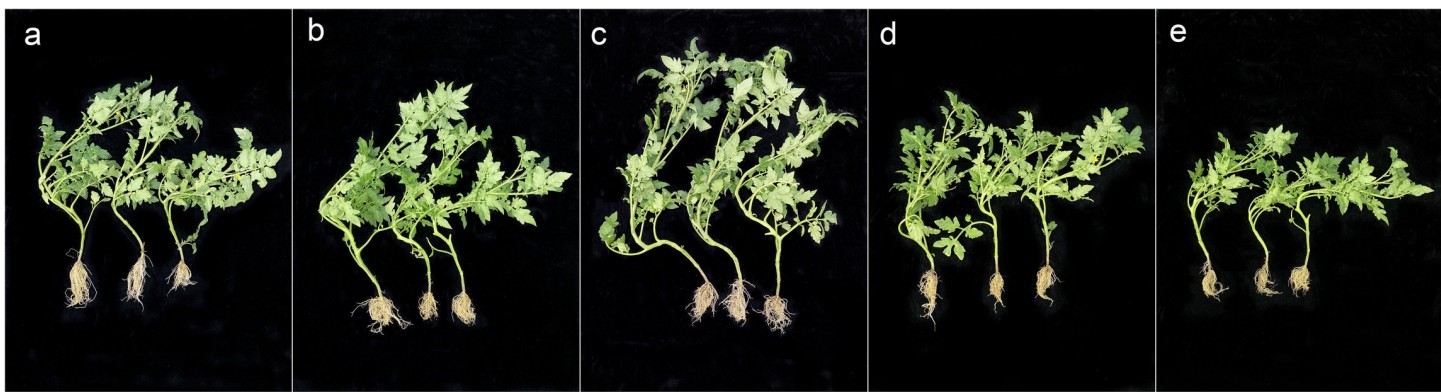

**Fig 10. The morphological indicators of tomato seedlings treated with different strain FX-6 dilutions.** (a-d) Treated with 10, 50, 100 and 200 fold-dilution. (e): Control.

result suggested that FX-6 may could produced antibacterial secondary metabolites and the cultivation time could affected antibiotic production. Consequently, the use of optimal growth conditions, such as time, is a simple and rapid method to improve the antagonistic potential toward phytopathogens. For instance, *P. polymyxa* JY1-5 produced the maximum yield of antimicrobial compound after 36 h of incubation in KL medium [1]. Our research results showed that *B. velezensis* FX-6 could promote tomato seed germination and seedling growth. However, undiluted FX-6 fermentation broth inhibited tomato seed germination (Fig 8), which could be associated with volatiles released from the bacterium. Volatiles can be a powerful tool for many PGPRs to combat with other microorganisms including plant pathogens [37]. Since the compounds are not target specific, plant growth can be affected as well. Such effects can be either promotion or inhibition for plant growth [38]. For example, *B. velezensis* BAC03 inhibited seed germination and seedling growth on eight tested plants, because the concentration of released chemicals were relatively high towards small seedlings, while the concentration decreased over time, and to be beneficial to promote growth when the plant approaches maturity [39]. Concentration of secreted chemicals of the volatiles released from various isolates could be responsible for these contrasting effects [40]. According to this speculation, volatiles released from FX-6 could not be deleterious to the plants at low concentrations, therefore, the FX-6 fermentation broth must be diluted when in use. Moreover, strain FX-6 could promote a significant increase in plant height, stem thickness, taproot length, and fresh weight of tomato seedlings (Figs 9 and 10). Previously, *Bacillus velezensis* WZ-37 also could promote a significant increase in plant height, stem diameter, fresh weight, and dry weight of tomato seedlings, however [41], its plant growth-promotion substances were not estimated.

As we know, bacterial plant growth promotion was a well-established and complex phenomenon, and was often achieved by the activities of more than one plant growth promoting trait induced by the associated bacterium [26]. FX-6 could produces IAA, siderophore, and has ACC deaminase activity (Fig 7). These compounds are all potentially associated with plant growth promotion as have been demonstrated by many researchers. IAA is the mainly naturally occurring auxin, and can be provided by PGPRs, or plants that are induced by certain PGPR [39]. Our results demonstrate that the isolate FX-6 strain could produce IAA, and l-tryptophan is not necessary for FX-6 to produce IAA. Siderophores (Greek: "iron carrier") are small, high-affinity iron chelating compounds secreted by microorganisms such as bacteria, fungi and grasses, which has high affinity with $Fe^{3+}$ [26]. Thus, the strain FX-6 could effectively prevent growth and reproduction of pathogens by binding the most utilizable $Fe^{3+}$ in the rhizosphere. The ACC deaminase of FX-6 promotes plant growth by lowering plant ethylene levels. This enzyme catalyzes the conversion of ACC, the immediate precursor of ethylene synthesis in plants, to ammonia and α-ketobutyrate [42]. In conclusion, *B. velezensis* strain FX-6 has activities of plant growth promotion through the secretion of several substances. While, the molecular mechanisms of *B. velezensis* promote plant growth are still unknown at present. So in the future, we will figuring out the molecular mechanism of *B. velezensis* FX-6 promoting plant growth.

## Conclusions

The isolated *B. velezensis* FX-6 showed remarkable plant biocontrol properties, which could effectively inhibit the development of tomato gray mould disease. FX-6 exhibits strong inhibitory activity against a variety of phytopathogenic fungi. When strain FX-6 was cultured for 72 h, the antagonistic effects against *B. cinerea* was optimal. Furthermore, FX-6 showed promote tomato seed germination and tomato plant growth activity, which was verifed in the tomato plant pot experiment. The above result indicated that strain FX-6 efficiently protected tomato

from *B. cinerea* infection and constituted a promising biocontrol agent against plant diseases induced by phytopathogenic fungi.

## Supporting information

**S1 File. 16S rDNA sequence of *B. velezensis* FX-6.**
(PDF)

**S2 File. gyrA sequence of *B. velezensis* FX-6.**
(PDF)

## Acknowledgments

We are grateful to Plant pathology laboratory in Institute of Plant Protection of Gansu Academy of Agricultural Sciences for providing the six plant pathogenic fungi: *Botrytis cinerea*, *Fusarium oxysporum, Sclerotinia sclerotiorum*, *Rhizoctonia solanikuhn*, *Colletotrichum gloeosporioides* and *Colletotrichum acutatum*.

## Author Contributions

**Conceptualization:** Zhaoyu Li, Jiajia Li, Tong Shen.

**Data curation:** Zhaoyu Li, Jiajia Li, Mei Yu.

**Formal analysis:** Zhaoyu Li, Jiajia Li, Tian Tian.

**Funding acquisition:** Zhaoyu Li, Tong Shen.

**Methodology:** Zhaoyu Li, Jiajia Li, Mei Yu.

**Project administration:** Tian Tian.

**Supervision:** Tian Tian.

**Writing – original draft:** Zhaoyu Li.

**Writing – review & editing:** Zhaoyu Li, Jiajia Li, Peter Quandahor, Tong Shen.

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
