## [Decision Letter · Decision Letter 0]

6 Mar 2023

PONE-D-22-25649Isolation and identification of *Bacillus velezensis*  FX-6 and its biocontrol activity on * Botrytis cinerea* and tomato plant growthPLOS ONE

Dear Dr. Li,

Thank you for submitting your manuscript to PLOS ONE. After careful consideration, we feel that it has merit but does not fully meet PLOS ONE’s publication criteria as it currently stands. Therefore, we invite you to submit a revised version of the manuscript that addresses the points raised during the review process.

Dear Author, Reviewer’s comments on the manuscript have now been received. Having considered these comments alongside your paper there is need of major revisions. If you feel that the manuscript could be significantly revised based on the reviewer’s comments, you could consider resubmitting the manuscript after through revisions.  See detailed comments and suggestions on your manuscripts received from the potential reviewers.

We look forward to receiving your revised manuscript.

Kind regards,

Raees Ahmed, Ph.D.

Academic Editor

PLOS ONE

Journal Requirements:

Reviewers' comments:

Reviewer's Responses to Questions

**Comments to the Author**

1. Is the manuscript technically sound, and do the data support the conclusions?

Reviewer #1: No

Reviewer #2: Yes

2. Has the statistical analysis been performed appropriately and rigorously? 

Reviewer #1: No

Reviewer #2: Yes

3. Have the authors made all data underlying the findings in their manuscript fully available?

Reviewer #1: Yes

Reviewer #2: Yes

4. Is the manuscript presented in an intelligible fashion and written in standard English?

Reviewer #1: No

Reviewer #2: No

5. Review Comments to the Author

Reviewer #1: Abstract The abstract of the manuscript is very poorly written with less clarity. I suggest improving it and make it technically strong. The promising results must be briefly explained in the abstract.

Introduction

Line 25 B. cinerea write its full name when using first time.

Line 29 - 30 This statement is not clear.

Line 37 (Army of attention) what does it mean?

Line 40 - 41 Statement is very confusing.

Line 49 - 52 This part is not clearly written.

Line 55 - 57 Add complete address of fungal collection Bank.

Line 58 - 60 Have you confirmed the pathogenicity of the collected fungal agents?

Line 55 - 64 The sentences are very confusing. Re-write them.

Line 72 - 73 Which fungal strains were used in this screening test?

Line 75 At what temperature the plates were incubated and for how many days?

Line 83 How bacterial characters were directly observed? At what lens power cell size and morphology was studied?

Line 101 - 105 Why was this test done for FX – 6 when screening test was already done? Furthermore, there is no information on the No. of replications and how inhibition zone was measured?

Line 106 - 115 The given information is very confusing. Why fermentation time finding related test was only performed against B. cinerea only?

Line 116 - 122 Was it a detached leave assay? Why only B. cinerea was selected for this study? Where were the leaves cultured? If it was a detached leaf assay, then how leaves were not wilted up to 5 days?

Line 144 What was the incubation period? It indicates that it has phosphorous solubility …….. Delete it.

Line 148 Write down the full name of DF salt minimal medium.

Line 152 - 160 How same sized tomato seeds were selected? Where were the treated seeds placed before measuring germination percentage etc.? What was the size of the pot used and quantity of the soil? Was it a replicated trial?

Line 166 - 174 Why only FX – 6 was selected for further studies whereas many other strains were almost equally effective against B. cinerea. Mention there antagonistic activity against other fungal isolates as well.

Line 183 - 188 The results are not properly explained. Why only FX – 6 was selected to test against other pathogenic fungal agents?

Line 189 Table 2 Inhibition rates of strain F3A against seven pathogenic fungi ….. It should be cross checked.

Line 199 - 206 The results are not properly explained. Thins are very confusing.

Line 218 - 223 The results are not properly explained.

Line 233 Table 4. Fresh weight（cm）…… Cross check the Unit. What is CK in the table?

Discussion Discussion portion is very poor. The results are not properly discussed with already published literature.

Line 280 The study does not reflect claim to identify a new bacteria agent as an antagonist. This objective is not mentioned anywhere in the manuscript.

Conclusion Conclusion needs to be more eye catching and logical.

Reviewer #2: I have completed my review of the submission “Isolation and identification of Bacillus velezensis FX-6 and its biocontrol activity 1 on Botrytis cinerea and tomato plant growth”. The authors have evaluated the effect of Bacillus strain on growth promotion of tomato plants. They have conducted several experiments to support their hypothesis. The study presents the results of original research and the results reported have not been published elsewhere. However, the manuscript needs improvements.

My comments on the manuscript are as under

1. The title can be modified as “Bacillus velezensis FX-6 suppresses the infection of Botrytis cinerea and increases the biomass of tomato plants”

2. Line 9: Replace “Botrytis cinerea-caused tomato gray” with “Botrytis cinerea causing tomato gray”

3. Line 11: While mentioning the code of the strain for 1st time please write the name of the species

4. Line 25: Replace “is one of the countries with the largest area of production worldwide” with “is one of the countries with the largest cultivated area worldwide”

5. Line 36-37: Replace “Bacillus strains have received an army of attention as biocontrol agents” with “Bacillus strains have received a huge attention as biocontrol agents”

6. Line 40-41: Replace “Up to now, there are just several reports on the control of tomato gray mold by B. velezensis” with “There are several reports on the control of tomato gray mold by B. velezensis”

7. It is better to replace the word biocontrol fungicide with biopesticide

8. Language needs improvement throughout the manuscript

9. The result representation is very weak. The tables should be replaced with high resolution graphs

10. Discussion must be improved and supported with more references and logical reasoning.

11. Manuscript needs formatting according to journal’s format

6. PLOS authors have the option to publish the peer review history of their article (what does this mean?). If published, this will include your full peer review and any attached files.

Reviewer #1: No

Reviewer #2: **Yes: **Muhammad Naveed Aslam

---

## [Author Response · Author response to Decision Letter 0]

27 Mar 2023

Reviewer #1（Comments）: 

The abstract of the manuscript is very poorly written with less clarity. I suggest improving it and make it technically strong. The promising results must be briefly explained in the abstract. 

Response: the abstract of the manuscript has been revised to make the content clarity. The promising results have been briefly explained in the abstract.

Line 25 B. cinerea write its full name when using first time.

Response: B. cinerea has been changed to Botrytis cinerea. 

Line 29 - 30 This statement is not clear. 

Response: this statement has been supplemented to make it clear, and added a reference.

Line 37 (Army of attention) what does it mean?

Response: an army of attention has been changed to a huge attention.

Line 40 - 41 Statement is very confusing.

Response: the content here has been revised to make the expression clear.

Line 49 - 52 This part is not clearly written.

Response: this part has been rewritten to express clearly.

Line 55 - 57 Add complete address of fungal collection Bank.

Response: complete address of fungal collection Bank already added.

Line 58 - 60 Have you confirmed the pathogenicity of the collected fungal agents?

Response: yes, I have confirmed the pathogenicity of the collected fungal agents.

Line 55 - 64 The sentences are very confusing. Re-write them.

Response: the sentences have been carefully rewritten to make them clear. 

Line 72 - 73 Which fungal strains were used in this screening test?

Response: the fungal strain used in this screening test was B. cinerea, and has been supplemented in the article.

Line 75 At what temperature the plates were incubated and for how many days?

Response: the plates were cultured at 25 ℃ for about 6 days, and has been supplemented in the article. 

Line 83 How bacterial characters were directly observed? At what lens power cell size and morphology was studied?

Response: a single colony was taken and spread evenly on a glass slide and then dry-fixed and Gram-stained for 1 min. Next, the morphology, cell size, and spore characteristics of strain FX-6 was observed under a microscope.

Line 101 - 105 Why was this test done for FX-6 when screening test was already done? Furthermore, there is no information on the No. of replications and how inhibition zone was measured?

Response: the test here is the antagonistic activity of FX-6 against other seven pathogenic fungi, so as to determine the inhibition spectrum of FX-6. The No. of replications is three. The inhibition zone is not measured, we calculated the inhibitory rate by measuring the colony diameter.

Line 106 - 115 The given information is very confusing. Why fermentation time finding related test was only performed against B. cinerea only?

Response: this study is aimed at screening biocontrol bacteria for tomato gray mold, so the fermentation time finding related test was only performed against B. cinerea.

Line 116 - 122 Was it a detached leave assay? Why only B. cinerea was selected for this study? Where were the leaves cultured? If it was a detached leaf assay, then how leaves were not wilted up to 5 days?

Response: yes, it was a detached leave assay. Because my study is aimed at screening biocontrol bacteria for tomato gray mold, so only B. cinerea was selected for this assay. The petioles of isolated leaves were wrapped with soaked cotton, and then placed in a petri dish with wet filter paper. The cotton balls and filter paper were moist during the whole test process, which ensured the leaves were not wilted up to 5 days.

Line 144 What was the incubation period? It indicates that it has phosphorous solubility …….. Delete it.

Response: the expression of the incubation period is incorrect and has been deleted. The sentence “It indicates that it has phosphorescent solubility...” has been deleted.

Line 148 Write down the full name of DF salt minimal medium.

Response: the full name of DF salt minimal medium is Dworkin and Foster (DF) salt minimal medium, and has been added.

Line 152 - 160 How same sized tomato seeds were selected? Where were the treated seeds placed before measuring germination percentage etc.? What was the size of the pot used and quantity of the soil? Was it a replicated trial?

Response: just look at the same size with eyes. The treated seeds were placed in a petri dish with wet filter. The diameter of pot was 20 cm, the height was 15 cm, and the quantity of soil was about 1.6 kg. It was a replicated trial, and each treatment was repeated three times. 

Line 166 - 174 Why only FX – 6 was selected for further studies whereas many other strains were almost equally effective against B. cinerea. Mention there antagonistic activity against other fungal isolates as well.

Response: compared with other strains, strain FX-6 has the highest antagonistic activity against B. cinerea, and the antagonistic activity did not significantly decrease from the first generation to the tenth generation. The antagonistic activity against other fungal isolates was the content of the inhibitory spectrum of strain FX-6 later, so it should not be mentioned here.

Line 183 - 188 The results are not properly explained. Why only FX – 6 was selected to test against other pathogenic fungal agents?

Response: because strain FX-6 has the highest antagonistic activity against B. cinerea, and the antagonistic activity did not significantly decrease from the first generation to the tenth generation. Therefore, only FX-6 was selected to test against other pathogenic fungal agents.

Line 189 Table 2 Inhibition rates of strain F3A against seven pathogenic fungi ….. It should be cross checked. 

Response: I have carefully checked and corrected the mistakes found.

Line 199 - 206 The results are not properly explained. Thins are very confusing.

Response: the results here have been revised.

Line 218 - 223 The results are not properly explained.

Response: the results here have been revised.

Line 233 Table 4. Fresh weight（cm）…… Cross check the Unit. What is CK in the table?

Response: the fresh weight unit is wrong and has been corrected. CK in the table is the seeds treated with water.

Discussion portion is very poor. The results are not properly discussed with already published literature.

Response: the discussion portion has been revised, and new references have been added.

Line 280 The study does not reflect claim to identify a new bacteria agent as an antagonist. This objective is not mentioned anywhere in the manuscript.

Response: this expression is wrong and has been corrected.

Conclusion needs to be more eye catching and logical.

Response: the conclusion has been revised.

Reviewer #2 (Comments): 

1.The title can be modified as “Bacillus velezensis FX-6 suppresses the infection of Botrytis cinerea and increases the biomass of tomato plants”

Response: the title has been modified as “Bacillus velezensis FX-6 suppresses the infection of Botrytis cinerea and increases the biomass of tomato plants”.

2. Line 9: Replace “Botrytis cinerea-caused tomato gray” with “Botrytis cinerea causing tomato gray”

Response: “Botrytis cinerea-caused tomato gray” has been changed to “Botrytis cinerea causing tomato gray”.

3. Line 11: While mentioning the code of the strain for 1st time please write the name of the species

Response: the species name Bacillus velezensis has been added before FX-6.

4. Line 25: Replace “is one of the countries with the largest area of production worldwide” with “is one of the countries with the largest cultivated area worldwide”

Response: “is one of the countries with the largest area of production worldwide” has been changed to “is one of the countries with the largest cultivated area worldwide”.

5. Line 36-37: Replace “Bacillus strains have received an army of attention as biocontrol agents” with “Bacillus strains have received a huge attention as biocontrol agents”

Response: “Bacillus strains have received an army of attention as biocontrol agents” has been changed to “Bacillus strains have received a huge attention as biocontrol agents”.

6. Line 40-41: Replace “Up to now, there are just several reports on the control of tomato gray mold by B. velezensis” with “There are several reports on the control of tomato gray mold by B. velezensis”

Response: “Up to now, there are just several reports on the control of tomato gray mold by B. velezensis” has been changed to “There are few reports on the control of tomato gray mold by B. velezensis”.

7. It is better to replace the word biocontrol fungicide with biopesticide

Response: the word biocontrol fungicide has been changed to biopesticide.

8. Language needs improvement throughout the manuscript

Response: the language of the manuscript has been improved.

9. The result representation is very weak. The tables should be replaced with high resolution graphs

Response: the result representation has been revised. The tables have been replaced with high resolution graphs.

10. Discussion must be improved and supported with more references and logical reasoning.

Response: The discussion has been revised and new references have been added.

11. Manuscript needs formatting according to journal’s format.

Response: Manuscript has been formatting according to journal’s format.

---

## [Decision Letter · Decision Letter 1]

29 May 2023

*Bacillus velezensis * FX-6 suppresses the infection of  * Botrytis cinerea* and increases the biomass of tomato plants

PONE-D-22-25649R1

Dear Dr. Li,

We’re pleased to inform you that your manuscript has been judged scientifically suitable for publication and will be formally accepted for publication once it meets all outstanding technical requirements.

Kind regards,

Durgesh Kumar Jaiswal, Ph.D.

Academic Editor

PLOS ONE

Additional Editor Comments (optional):

Thank you for successfully, address the all reviewers comments and improved the manuscript. therefore, I have recommended the manuscript for publication

Reviewers' comments:

Reviewer's Responses to Questions

**Comments to the Author**

1. If the authors have adequately addressed your comments raised in a previous round of review and you feel that this manuscript is now acceptable for publication, you may indicate that here to bypass the “Comments to the Author” section, enter your conflict of interest statement in the “Confidential to Editor” section, and submit your "Accept" recommendation.

Reviewer #2: All comments have been addressed

2. Is the manuscript technically sound, and do the data support the conclusions?

Reviewer #2: Yes

3. Has the statistical analysis been performed appropriately and rigorously? 

Reviewer #2: Yes

4. Have the authors made all data underlying the findings in their manuscript fully available?

Reviewer #2: Yes

5. Is the manuscript presented in an intelligible fashion and written in standard English?

Reviewer #2: Yes

6. Review Comments to the Author

Reviewer #2: (No Response)

7. PLOS authors have the option to publish the peer review history of their article (what does this mean?). If published, this will include your full peer review and any attached files.

Reviewer #2: No

---

## [Editor Report · Acceptance letter]

4 Jun 2023

PONE-D-22-25649R1 

*Bacillus velezensis* FX-6 suppresses the infection of *Botrytis cinerea* and increases the biomass of tomato plants 

Dear Dr. Li:

I'm pleased to inform you that your manuscript has been deemed suitable for publication in PLOS ONE. Congratulations! Your manuscript is now with our production department. 

Kind regards, 

on behalf of

Dr. Durgesh Kumar Jaiswal 

Academic Editor

PLOS ONE